# Do Consumers Value Welfare and Environmental Attributes in Egg Production Similarly in Fresh Eggs and Prepared Meals?

**DOI:** 10.3390/ani13030324

**Published:** 2023-01-17

**Authors:** Maurice Doyon, Stéphane Bergeron, Laure Saulais, Marie-Ève Labonté, Véronique Provencher

**Affiliations:** 1Agricultural Economics and Consumer Science, Laval University, Quebec, QC G1V 0A6, Canada; 2School of Nutrition, Laval University, Quebec, QC G1V 0A6, Canada

**Keywords:** poultry, welfare, eggs, consumer choice, sustainable production

## Abstract

**Simple Summary:**

Using a stated preference survey, this study examines consumer preferences for eggs from four production systems. Choices of type of eggs for fresh eggs are compared with choices of two prepared meals containing eggs. Additionally, choices are examined before and after an information treatment based on scores expressing egg production systems’ animal welfare, nutrition, or environmental impact. Results indicate that respondents choose eggs from more sustainable production systems more frequently when included in prepared meals compared to fresh eggs. Information conveyed in scores led participants to update prior beliefs and revise their initial choices, especially for animal welfare attributes.

**Abstract:**

Food items are increasingly chosen based on sustainability attributes as the public is becoming increasingly aware of the environmental and animal welfare impacts of production systems, in addition to the traditional consideration for nutrition. Although surveys have been used to investigate the demand for these attributes in unprocessed products, little information exists on how these attributes impact consumer preferences in the case of processed products or prepared meals. This study uses a stated preference survey to examine Quebec (Canada) consumers’ preferences for eggs from four production systems with different impacts on animal welfare and on the environment. We compare the respondents’ choices of fresh eggs and two prepared meals that contain eggs. Furthermore, we examine the shift in choices following information treatments on animal welfare, nutrition, or environmental impact attributes. Results indicate that respondents choose eggs from more sustainable production systems more frequently when included in prepared meals than in their unprocessed form. The provision of information led participants to update prior beliefs and revise their initial choices, especially for animal welfare attributes.

## 1. Introduction

Consumers’ food quality perception results from a complex integration of product-level quality attributes (such as sensory attributes and nutritional properties), as well as process-level quality attributes such as production and processing attributes [1]. In North America, process-level quality attributes in food are gaining in importance, as reflected by the forecast annual growth of more than 6% in sustainable foods for the period 2021–2028 [2]. This growth responds to an increase in consumers’ concerns regarding the social and ethical values of food production systems. For instance, surveys show that consumers report being increasingly sensitive to farm animal welfare (FAW) and to the environmental impacts of livestock production [3,4,5,6], even though these stated preferences results are not always reflected in their purchasing behaviors [7].

Although there is an abundance of papers estimating consumer preference for social attributes in food, it appears that little attention has been paid to the value of these same attributes in value-added goods such as processed foods and prepared meals. As an illustration, a large-scale meta-analysis on stated preference [8] for FAW focuses on animal products in an unprocessed or minimally processed form, such as cuts of meat, milk, or fresh eggs. The exclusion of processed animal products was not part of the selection criteria for the meta-analysis, but the outcome was rather representative of the studies available.

The lack of studies on consumer preferences for FAW in processed products is surprising, considering the increased consumption of prepared meals observed in many countries [9,10] as well as increasing household expenditure for food consumed out-of-home [11]. Hence, a closer examination of consumer preference regarding social attributes in transformed products is warranted, to determine whether consumer preferences for social attributes change when going from a basic product to a prepared meal.

Another area that has been overlooked in the food preference literature is the conflicting interaction between social and environmental attributes that can sometimes arise. Currently, most research focuses on a single social (e.g., healthiness or animal welfare impact) or environmental (e.g., carbon footprint) attribute or considers attributes additively. Few studies examine conflicting attributes. Nevertheless, [12] examine how organic products competed for the same customers as animal welfare-oriented products, while [13] studied both animal welfare and environmental attributes for eggs produced under different systems (caged, free run, free range, organic). In the latter, the attributes were considered as providing added and complementary benefits. Yet, in many instances, attributes may in fact counteract each other. As an illustration, [14] show that GHG emissions per kilogram of eggs are greater for organic and free-range egg production than for battery cage eggs. Thus, consumers concerned with animal welfare and environmental attributes must make a trade-off when purchasing these products.

To address these issues, this study uses an online survey with choice tasks for eggs produced in different housing systems (cage, enriched housing, free run, and free range) which can be associated with different levels of animal welfare and environmental impacts. This allows for an examination of the consistency of consumer preferences for type of eggs across the following two axes: (i) when going from the unprocessed product to the same product used as an ingredient in a prepared meal, and (ii) when additional information is provided that may update their beliefs or require trade-offs. Specifically, the first part of the study examines preferences regarding type of eggs and if these preferences switch when going from a dozen eggs to prepared meals that include eggs. The second part of the study uses the same products but examines stability of choices after providing additional information regarding one of the following attributes: animal welfare, nutrition, or environmental impact. The information is provided using a color-coded score that provides a quick assessment of the relative rating of each product.

## 2. Materials and Methods

### 2.1. Experimental Products

The products used in this study are fresh eggs and prepared meals that include eggs as ingredients. Eggs have several interesting characteristics suitable for our experimental design. First, they can be purchased fresh and are also found in prepared meals. Furthermore, eggs are readily available from a variety of production (housing) systems in Canada, that have different outcomes in terms of social impact (animal welfare) and environmental impact (carbon footprint). Note that the nutritional value of eggs is maintained constant in our experiment since the housing system does not affect nutritional value (Housing systems can affect some parameters of eggs such as whipping capacity and shell resistance. However, macronutrients (fats, proteins and carbohydrates) familiar to consumers are not affected by housing systems [15]). This study considers the following four types of egg: conventional cage (cage) that restricts birds’ movement to a small space; the enriched housing system (enriched) which houses birds in a colony of up to 200 birds with more space per bird compared to conventional cage, allows movement and provides furnishing such as perches, nest boxes, and scratching pads; the free-run production system (free run) which houses birds in large common areas (floor or aviary), with access to nesting boxes and perches; the free-range production system (free range), which is similar to the free-run system, but with the added benefit of access to the outdoors when permitted by weather conditions. For this paper, we will use *type of eggs* to refer to the four production systems previously described.

The four types of eggs are naturally associated with FAW properties. Information about hen welfare has been discussed in the media over the last decade, with several fast-food chains as well as food processors and retailers in the United States and Canada committing to sell free-run or free-range eggs for animal welfare reasons [16]. Meanwhile, a body of research suggests that enriched housing offers numerous advantages to hens’ health and well-being, even when compared to free-run or free-range production [17,18,19], and better aligns with consumer expectations of FAW production [20,21]. However, few consumers are aware of this production system and its advantages for hens [20].

The information available in the press and the announcement made by large food companies or retailers combined with a lack of technical knowledge about egg production and housing system, likely result in a majority of individuals believing that free-run and free-range systems offer greater FAW than enriched housing. This is also confirmed by results in [20]. This provides an ideal setting for an information treatment that presents information that differs from common beliefs and that is therefore susceptible to impact the choice of type of eggs based on FAW. Thus, based on the literature as well as for experimental design purpose, we classified FAW by increasing order as follows: cage, free-run, free-range, and enriched housing.

Regarding environmental impacts, production systems can be regrouped. In the low impact group, we have cage and enriched housing while in the slightly higher impact one we have free run and free range [17].

### 2.2. Overview of Study Design

An online questionnaire was developed to explore the stability of consumer choices regarding the type of eggs across the possible choices and how information about social and environmental attributes affects these choices. The questionnaire contained two series of choice tasks. In each choice task, respondents had to select their preferred option among the four types of eggs for three products: a dozen eggs, a spinach salad topped with slices of hard-boiled egg and a vegetable quiche. A choice task was first provided without social or environmental information, and then presented a second time with one type of information. Note that independently of the information treatment, participants were provided with a short description of each housing system at the beginning of the survey. Figure 1 illustrates the choice set used in the survey for the quiche by type of eggs. The order of presentation was determined randomly within each choice task.

The structure of the questionnaire is as follows (questionnaire available in Appendix A): 1—participants were asked to answer two inclusion questions to verify whether they consumed eggs and were involved, at least in part, in household food purchases; 2—the eligible participants were then randomly allocated to one price set (PS1 or PS2; described below); 3—participated in three choice tasks, one for each of the following product: fresh eggs, quiche, salad with hard eggs; 4—in the second step, participants were randomly allocated to one out of three information treatments that addressed one of the following topics: nutrition, animal welfare, or environmental impact; 5—the information for each topic was presented using a synthetic score; 6—after explanation on the scoring system, respondents repeated their same three choice tasks seen prior to the information, but this time with the color-coded scores included beneath each product. Figure 2 provides a graphical representation of the flow of the questionnaire.

### 2.3. Price Structures

The prices of the egg products increased according to the type of egg production system in the following ascending order; cage, enriched, free run, and free range. This progression reflects retail prices in Canada. All participants saw the following prices for the different types of dozen eggs: CAD 3.59 for cage, CAD 4.10 for enriched, CAD 5.89 for free run, and CAD 6.10 for free range (Table 1). These prices are average market prices collected in Quebec City (Canada) at the time of the study. The egg component of the salad and quiche could be from one of the four housing systems previously described. Given that the market does not currently make this distinction, two price scenarios (PS1 and PS2) were created. In PS1, the price differential between the types of eggs is HIGH for the quiche and LOW for the salad. The opposite applies in PS2, as indicated in Table 1. Note that the LOW price treatment offers value-added eggs at little cost. To illustrate, the cost of going from a quiche made with cage eggs to one made with free-range eggs is CAD 2.00 in the HIGH price treatment, while the same movement costs CAD 0.30 in the LOW price treatment.

### 2.4. Information Treatments and Scores

The goal of the second part of the questionnaire was to examine the effect of additional information on consumers’ preference for types of eggs across products. The information provision was about animal welfare or environmental impact or nutritional qualities. The format used was that of a label indicating a synthetic score. It is based on the Nutri-Score system, which was developed in France as an indicator of nutritional quality and displayed on front-of-pack labels [22]. Nutri-Score provides a single assessment of the overall nutritional quality of food and beverages and has been shown to be quick to process and easy to understand by most consumers [23,24]. Visually, the scoring system is a scale of four colors, with the best scoring food items shown in dark green, followed with yellow, orange, and red (the lowest score). These colors are also combined with a letter ranging from A to E, allowing comparison of products within a given product category. In the current study, three similar scoring systems were developed to signal and compare the four types of eggs according to their level of animal welfare, their environmental impact, or their nutritional quality, with a scale from A to E.

The participants were briefly introduced to the scoring systems and provided with information on criteria used to calculate the scores. The following paragraph provides an overview of what was presented to the participants. The full text is part of the survey and available in the Appendix A.

Animal Welfare—Each type of housing system was assessed according to a list of criteria found in the appropriate literature, including the following: ability to express natural behavior, health observation (fractures, pecking), hygiene criteria, health safety, and air quality.

Nutrition—Nutrient profiling was used to characterize the overall nutritional quality of the product by taking into account its content in several nutrients or ingredients of public health interest (e.g., sugars, protein, sodium, fat, fiber, whole grains). We calculated the scores for eggs and with consideration to all the other ingredients in the prepared meal products.

Environment—The impact of the production method of each ingredient was evaluated using life-cycle analysis (LCA) found in the appropriate literature, based on the proportion of ingredients in each product and relative to its product group. The score provides a global vision considering greenhouse gas emissions and other environmental impacts.

In all treatments, it was stated that the score was calculated according to methods developed and validated by experts in the field. The resulting grading for all products is illustrated in Table 2. Note that some changes in score occur along the product axis (quiche vs. salad) and for others, it is across types of eggs axis (cage vs. free range). The choice task with the information treatment was identical to the one illustrated in Figure 1, with as a difference an additional score line beneath the price stating the determined score letter and its corresponding color.

### 2.5. Recruitment Procedure

The survey was approved by Laval University Ethics Committee (2019-069) and conducted by a professional firm through their online panel that is representative of the population in the province of Quebec (Canada). Participants were recruited via e-mail in the first week of October 2019. The three following criteria determined eligibility: (i) age of majority (min. 18 years old); (ii) has a role in household food purchases; and (iii) consumption of fresh eggs at least once a month. Every eligible respondent who participated received compensation from the firm in the form of reward points that could be converted into shopping gift cards.

### 2.6. Analysis

The focus of this paper is to examine individual choices and how they shift along our two axes of interest (level of processing and information).

Three possibilities exist when comparing a respondent’s choice for a prepared meal with the choice made for fresh eggs: (i) the selection of type of eggs is unchanged; (ii) the selection of type of eggs in the prepared meal is more expensive; (iii) the selection of type of eggs in the prepared meal is less expensive.

Similarly, after each information treatment a participant can for each product: (i) keep the selected type of eggs unchanged; (ii) select a more expensive type of eggs; (iii) select a less expensive type of eggs.

In both cases, percentages are calculated with regards to possible shifts across choices. When suitable, a chi-squared test is used to assess whether a difference in choices is statistically different at 5% or less (*p*-value < 0.05).

However, to further analyzed the data, the trajectory of shifts in choice also needs to be considered. As an illustration, a shift to cage eggs from enriched housing is not the same as a shift to cage eggs from free-range eggs. Some information is lost if we look only in aggregate at shifts to cage eggs without considering the point of departure. On the other hand, given that numerous possible trajectories exist, especially when considering information treatments, some results are presented by regrouping certain trajectories that carry similar information. For example, if several shifts in choices reduce cost while also improving the attribute of interest, they will group together for the purpose of the analysis.

## 3. Results

### 3.1. Sample Characteristics

Our sample consists of 905 participants that completed the questionnaires, nearly equally split between males (52%) and females (48%), with most participants (46%) between the age of 35 and 54 (Table 3). Over 60% of the sample declared being the main buyer of food for the household, and over 80% consumed more than a dozen eggs per month. The size of subsamples by information treatment is 303 for animal welfare, 303 for the environment, and 299 for nutrition.

### 3.2. Consistency of Type of Egg Preference in Fresh Eggs versus Prepared Meals

We first examined the choice of type of eggs as one moves from fresh eggs to prepared meals. As a baseline for comparison, the distribution of consumption for fresh eggs according to the four types of eggs is provided in Table 4. The most popular system for a dozen eggs was caged eggs, chosen by 41% (370) of the participants, followed by 24% (220) for enriched, 24% (217) for free run, and 11% (98) for free range.

Shifts in the type of eggs (same, more expensive, less expensive) from fresh eggs to prepared meals are reported in Table 5 by price scenario (HIGH and LOW differential). To calculate the percentages of individuals selecting a less expensive choice, we do not consider individuals who had initially chosen a dozen eggs from hens in cages, given that one cannot choose a less expensive type of egg from cage-type eggs. Similarly, to calculate the percentage of the more expensive type of eggs, we removed individuals who selected free-range fresh eggs since one cannot move up from the most expensive type of eggs. We first test the null hypothesis that the LOW and HIGH price treatments provide the same result in choice shifts when going from eggs to quiche or when going from eggs to salad. For both shifts, we reject the null hypothesis (Pearson’s chi-squared test; *p*-value < 0.05). Thus, price treatments affect choice shifts from eggs to quiche and from eggs to salad.

Results indicate that over half the participants maintained the same choice of eggs in the LOW price treatment, with 56% choosing the same type of egg for the quiche and 52% for the salad (Table 4). In the HIGH price treatment, where it is more costly to switch to a more expensive type of eggs in the prepared meal, a higher frequency of participants keeping the same choice is observed. Specifically, 71% kept the same types of eggs for the quiche and 69% for the salad. The proportion of participants who choose a more expensive type of egg is higher in the LOW price treatment compared to the HIGH one. For the quiche, 43% of participants chose a more expensive type of egg in the LOW treatment, compared to 22% in the HIGH treatment. As for the salad, the switch to a more expensive type of egg was 58% versus 23% in the LOW and HIGH treatments, respectively.

Similarly, one should expect to see more movement toward the less expensive type of eggs in the HIGH price treatment given that it represents more savings. This logic does not seem to hold for the quiche where the percentages of participants moving to the less expensive type of eggs are very similar across price scenarios at 15% (LOW) and 16% (HIGH). For the salad, however, a difference between price treatments is observed, with 7% going to less expensive type of eggs for the LOW price treatment and 17% going to a less expensive type of eggs in the HIGH price treatments.

### 3.3. Information Treatment and Choices

We first concentrate on the effects of the information treatment on the choice of type of eggs for fresh eggs. Table 6 and Figure 3A indicate that the information treatment on FAW has the largest effect, increasing the selection of enriched-type eggs from 25% to 58% and decreasing the selection for all the other types of eggs. Cage eggs registered the largest decrease in the selection, from 41% to 21%. The impact of the information treatment is statistically significant (Pearson’s chi-squared test *p*-value < 0.05). As indicated in Table 2, enriched housing has, in our information treatment, the best FAW score, while cage has the worst one.

Although the environmental score shows less movement in egg choice, its effect is statistically significant (Pearson’s chi-squared test *p*-value < 0.05). The score suggested that cage and enriched-type eggs were superior with regard to free-run and free-range types of eggs (Table 2). It also shows more stability in the choice of type of eggs (Table 6 and Figure 3B). The largest movement in choices is an increase from 24% to 38% for enriched-type eggs, while free-run-type eggs record a decrease in selection from 22% before the information treatment to 13% after.

The nutrition score was the same across all types of eggs (Table 2). Not surprisingly, revealing this information had little effect on choices, as indicated by Table 6 and Figure 3C. This is confirmed by a Pearson’s chi-squared test that failed to reject the null hypothesis that the choices are the same before and after the information treatment (*p*-value > 0.05). Nevertheless, Table 5 indicates a decrease in the selection of enriched eggs from 27% to 19% after the nutritional score was provided, with gains distributed to cage eggs and free-run eggs.

#### 3.3.1. Choice Shifts between Types of Eggs, By-Products, after Nutrition Information Treatment

Nutrition information in the form of color-coded scores did not generate significant changes in choice between types of eggs for the three products under study. More specifically, after seeing the nutrition score, which was identical across the four types of eggs, 80% of participants kept the same type of eggs when choosing fresh eggs (Table 7). Similarly, 81% and 77% did not change the type of eggs used in the quiche and the salad, respectively, after the information treatment. This behavior is found to be consistent across products (Pearson’s chi-squared *p*-value > 0.05). This is not surprising given that according to our score, the type of housing system has no impact on eggs’ nutritive value and thus on fresh eggs, the salad, and the quiche. In fact, this treatment isolates the price effects. Most of the respondents that modified their initial choice, switched to a less expensive option. This is most notable for the quiche, with 21%among those not in the least expensive option (cage) going to a less expensive option, compared to 6% among those not in the most expensive (free range) who switched to more expensive type of eggs.

#### 3.3.2. Choice Shifts between Types of Eggs, By-Products, after Animal Welfare Information Treatment

Table 8 presents the impact of using a score for FAW on choice shifts between types of eggs and by-products. The group size in the table represents the number of individuals who could make the specified shift. For example, in the ‘*No change (excluding enriched)*’ row, the group size is 240 for quiche. This group size represents the number of respondents in the FAW treatment that did not choose a quiche with enriched eggs as their initial choice before the FAW treatment. Among these 240 individuals, 105 (44%) did not change their initial choices of types of eggs for the quiche after seeing the FAW score.

Given that, based on the information that we provided, enriched housing scores best among the four types of eggs (Table 2), results in Table 8 are presented relative to enriched. Table 8 indicates that roughly half the participants did not change their choice after seeing the animal welfare score (*Total no change*). Specifically, 57% did not change their initial choice for fresh eggs, 51% for the quiche, and 49% for the salad (Table 8). However, these percentages include respondents that were already at the best animal welfare choice (enriched according to our scoring). Taking this into account, we excluded those that had chosen enriched before the information treatment (the *No Change (excl. enriched)* line). We then found that, after being shown that a better option for FAW existed than their initial choice, 47%, 44%, and 44% of participants kept their initial choice for fresh eggs, the quiche, and the salad, respectively. The distribution of behaviors was found to be the same across all products (Pearson’s chi-squared *p*-value > 0.05).

According to our scoring system, transitions to enriched-type eggs from any other housing system increases animal welfare. Shifting from cage to enriched provides the most FAW gain, based on our scoring, but also increases the monetary cost. Table 8 indicates that 45% of participants shifted from cage to enriched types (*To enriched with cost* row) for fresh eggs, 44% for the quiche, and 39% for the salad. Shifting from free run and free range to enriched not only brings a gain in animal welfare but also savings (*To enriched with savings* row). Table 8 shows that 45% shifted from free run or free range to enriched for fresh eggs, 51% did the same for the quiche, and 52% for the salad. The total percentage change from an initial choice other than enriched to enriched (with cost and with savings) is 45% for fresh eggs, 48% for the quiche, and 47% for the salad. It is interesting to note that the percentage of shifts enriched with savings is the same as those with the cost of a dozen eggs. However, shifts to enriched with savings for quiche and salad are more important than shifts with cost.

While most shifts occurred toward enriched, some did not. We regrouped these shifts under the ‘Shifts not toward enriched’ section in Table 8. For instance, after viewing the score on FAW, some participants changed their initial choice from enriched to cage or from free run and free range to cage, resulting in a reduction in FAW but with monetary savings (*Saving only* row). Percentages of this observed behavior are low, ranging from 2% for a dozen eggs to 6% and 4% for the quiche and the salad, respectively. One can argue that this behavior is mostly driven by monetary savings and low concerns for FAW in egg production.

Other types of change in choices include shifts to FAW gains at a higher cost, such as from cage to free run, cage to free range, and free run to free range. These shifts (*FAW gain at higher cost* in Table 8) occurrences are between 8% and 9% for all products. One possible explanation would be that one attribute of FAW, such as the free movement of the hens, is more important for these participants.

Finally, some participants made choices that appeared irrational by moving to a more expensive type of eggs with lower FAW (*Lost of FAW with cost* in Table 8). This includes any shift from enriched to free run or free range (In consumer economics, an irrational choice is changing for a basket of goods or services that offer less (in this case, lower score) at a higher cost. One might correctly argue that our scoring system for FAW might not be unanimously accepted by participants. However, here, the initial choice of participants exhibited an «irrational» choice before the information was enriched.). Table 8 indicates that among participants in a position to make such shifts from their initial choice of types of eggs (enriched), 6% did it for fresh eggs and the quiche and 15% for the salad. Although the percentage might appear important, they involved few individuals (Table 8) and might be the result of mistakes or misunderstandings of the choice task.

#### 3.3.3. Choice Shifts between Types of Eggs, By-Products, after Environmental Information Treatment

As indicated in Table 2, based on our scoring, the cage, and enriched systems obtain a slightly better environmental score than free run and free-range systems. After seeing these scores, a majority of individuals did not change their choice, with 67% choosing the same type of eggs for a dozen eggs, 62% for the quiche, and 61% for the salad (*no change* line in Table 9). However, a majority were already at the best environmental score (ES) following their initial choice. If we consider only participants whose initial choice can be improved, 44%, 45%, and 41% did not change their initial choice for fresh eggs, the quiche, and the salad, respectively (*Initial is not the best ES-no change* line in Table 9). The distribution of behaviors is the same across all products (Pearson’s chi-squared *p*-value > 0.05).

Transitions to a better score and less expensive choice are possible for the participants who chose free run or free range before the information treatment and shifted to either cage or enriched after the environmental scores. Such changes in choices were made by 42% of individuals for fresh eggs, 47% for the quiche, and 47% for the salad (*Better ES with saving*
Table 9).

It was also possible to shift to a type of egg that has the same ES but is less expensive. These shifts are from enriched to cage or from free range to free run. Among the participants in a position to make such shifts, 19%, 16%, and 18% did it for fresh eggs, the quiche, and the salad, respectively (*Same ES with saving* in Table 9).

Individuals who shift from their initial choice to a more expensive type of egg but with the same ES is 12% for fresh eggs and 9% for the quiche as well as for the salad (*Same ES with cost* in Table 9).

Finally, a minority of individuals shifted to a product that was more expensive and with a lower environmental score. These shifts are from cage or enriched eggs as an initial choice to free-run or free-range eggs after seeing the environmental score. These shifts are irrational from an economic point of view since the shift is to a lower score with a higher cost. These shifts represent 6% of the participants that were in a position to make such shifts for fresh eggs and the salad and 10% for the quiche.

## 4. Discussion

### 4.1. Social and Environmental Attributes Valuation in Fresh Eggs versus Eggs in Processed or Prepared Meal

Most participants maintained the same choice of type of eggs regardless of the degree of processing. When a difference in the choice is observed, the most common change is toward a more expensive type of egg in the prepared meal in comparison to fresh eggs. This tendency to upgrade the type of eggs is observed even when the price differential is high, but the number of individuals who upgrade doubles when price differentials are low. This result is somewhat surprising since one could expect that eggs are less salient when they are part of a prepared meal and, therefore, could be less valued. One assumption to explain the observed behavior is that consumers may have estimated the extra cost (and the added value) implied by having specialty eggs in the processed products as relative to the total cost (and value) of the food product. For example, a free-range dozen eggs is 70% more expensive than a dozen cage eggs. For the quiche in the HIGH price treatment, the same movement from cage-type eggs to free-range-type eggs results in a 29% increase in price. The relative price increase is, therefore, lower for the quiche and the salad.

Conversely, a minority of consumers (less than 17%) chose a less expensive type of egg in both the HIGH and LOW treatments. For these participants, the attributes associated with the type of eggs lose value when included in prepared meals.

### 4.2. Choice Changes after Information Treatment

In all treatments, most differences in choice after the information treatment suggest a rational response to the scores provided (improvement in attribute at no cost or with savings). Most individual shifts are in line with the information provided by the scores. This suggests that the visual scoring method is efficient in the way that it informs consumers and that it can influence their choices. In all treatments, a small group of individuals shifted toward choices that did not seem to respond logically to the information provided, e.g., shifting to a more expensive type of egg with a poorer score. These responses can represent individuals who have inconsistent preferences, who misunderstood the interpretation of the scores, or who filled out the survey carelessly. Nevertheless, the possibility that these choices were informed and deliberate cannot be ruled out. This could indicate, for instance, that some individuals based their decisions on other heuristics to ease the cognitive burden when trade-offs have to be made (for example, between FAW and the environment) or that they respond to the information on the basis of other undisclosed personal motives.

#### 4.2.1. Nutritional Information Effects

Unsurprisingly, the nutritional treatment produced the least amount of change in choices. In this treatment, according to the scoring system presented to participants, the type of eggs does not change the nutritional profile of eggs. Given that the nutritional scores remain the same regardless of the type of eggs, one could have expected significant changes to a less expensive option if nutrition was the dominating attribute in choice. Although some switches to a less expensive option are observed, the numbers are rather modest, with 19%, 21%, and 22% of the participants changing to a lower price egg for the dozen fresh eggs, the quiche, and the salad, respectively. Thus, bringing attention to nutrition by providing nutritional scores did not influence most individuals. This confirms that attributes other than nutrition and price are impacting consumers’ choices.

#### 4.2.2. Animal Welfare Information Effects

The information treatment with the most changes from the initial choice is the animal welfare information treatment. In this treatment, the scores informed that the enriched housing system has better FAW than both free-run and free-range systems. It is therefore not surprising that most people who had chosen eggs from the enriched housing system remained in this choice (93%). In addition, this information treatment likely updated the beliefs of some individuals who believed that free-range and free-run housing systems were offering the best level of FAW. The information (score) prompted them to change their choices to the less expensive enriched housing. Similarly, individuals who switched from cage to enriched may have been motivated by the low cost of obtaining the best option for the hen, according to our scores. In fact, most switches observed in this treatment are toward enriched housing, with the proportion of change of type of eggs being equal for those for whom this would incur at a cost (from cage) as from those who would see a saving (from free run and free range). This not only reveals that consumers are willing to pay for FAW when it is an affordable option, but it also reveals that using the simple scoring method was enough to inform individuals that reverse their initial choices made on prior beliefs regarding free run and free-range eggs.

Overall, significant shifts in choices are observed in this information treatment, suggesting that current markets might provide insufficient information to consumers. A minority of individuals who initially selected free range and free run did not change their choices. Possible explanations are that they value more specific aspects of FAW that they associate with these housing systems, such as totally free movements, or that they do not believe in the validity of the provided score. Food retailers, fast-food chains, and some processors have committed to free-run or free-range eggs under the pressure of animal welfare groups, possibly creating a validation for these types of eggs as the superior choice.

#### 4.2.3. Environmental Information Effects

In the environmental information treatment, recall that respondents only saw the environmental scores (ES) and not the FAW. Hence, we may assume that participants who selected free-range- and free-run-type eggs were motivated by some FAW reasons. However, when savings are combined with an increase in ES, more than 40% of participants that could make the shift from their original choice to one with a better environmental score did. More specifically, they shift from free-range or free-run type to cage or enriched. The results suggest a trade-off between environmental attributes and FAW, with over 40% of the respondents choosing the more ecological option at the cost of FAW. In this case, one must note that the trade-off also implies monetary compensation in the form of a lower market price.

## 5. Conclusions

The majority of respondents who selected different types of eggs for fresh eggs versus eggs in prepared meals selected a type of egg with higher FAW or environmental attributes in the prepared product. This has industry implications for expanding the use of eggs with social attributes in prepared meals.

Consumers make multiple food decisions daily, and the increasing array of products offered with social and environmental attributes may contribute to complexifying the cognitive burden of consumers as well as the ability of retailers to accurately predict market demand. Gaining an understanding of how these new dimensions of food preferences interact with other criteria and if a scoring system can reduce the cognitive burden of consumers, especially in the context of processed foods, seems important, considering the growing demand for these products.

Furthermore, social attributes can be competing, and the information that becomes common knowledge might not be based on scientific evidence. The case of FAW associated with types of production systems offered an interesting opportunity to test how information can impact consumer choices and reduce their cognitive burden. We found that a color code score can update prior beliefs and incite some individuals to change their choices. However, our study did not test the effect of providing information simultaneously on several social and environmental dimensions. In contrast, Canada’s strategy regarding front-of-package nutrition labeling has recently been updated to require a front-of-package nutrition symbol on foods with high sodium, sugars, and/or saturated fats [25]. There are currently no mandatory environmental or FAW labels on Canadian food markets, leading to multiple possible claims and formats. Further studies are needed to explore consumers’ responses to these multiple front-of-package labeling systems for different social and environmental attributes and the nature of the trade-offs made between attributes. Given that food items are a recurrent purchase, while we measure a single choice, it would be of interest to look at repeated purchase choices in future studies.

Moreover, our study used eggs in a hypothetical experimental setting, limiting the generalization of our results. Studies with a larger range of products in market settings with real transactions would be the next logical step.

## Figures and Tables

**Figure 1 animals-13-00324-f001:**
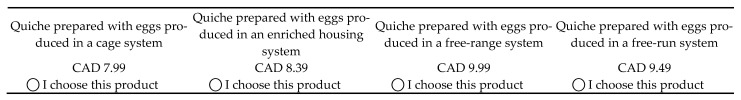
Choice set used in the online questionnaire for quiche.

**Figure 2 animals-13-00324-f002:**
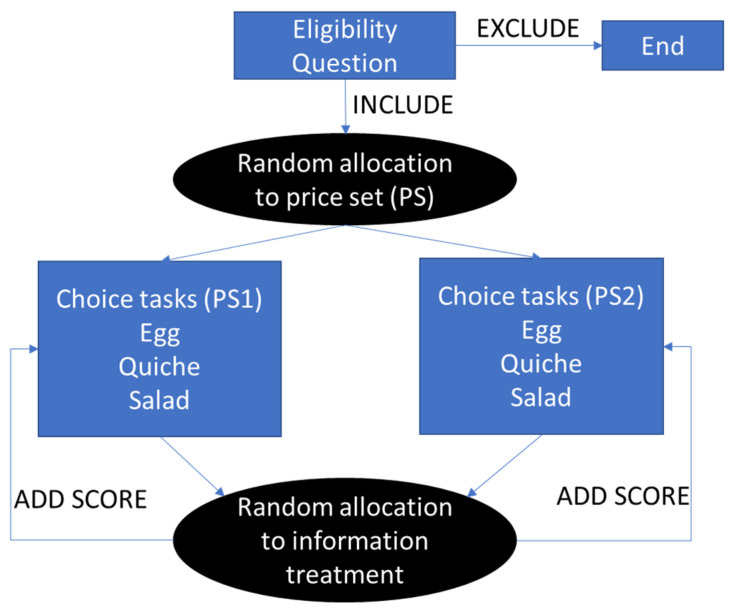
Experimental design illustrating the survey structure.

**Figure 3 animals-13-00324-f003:**
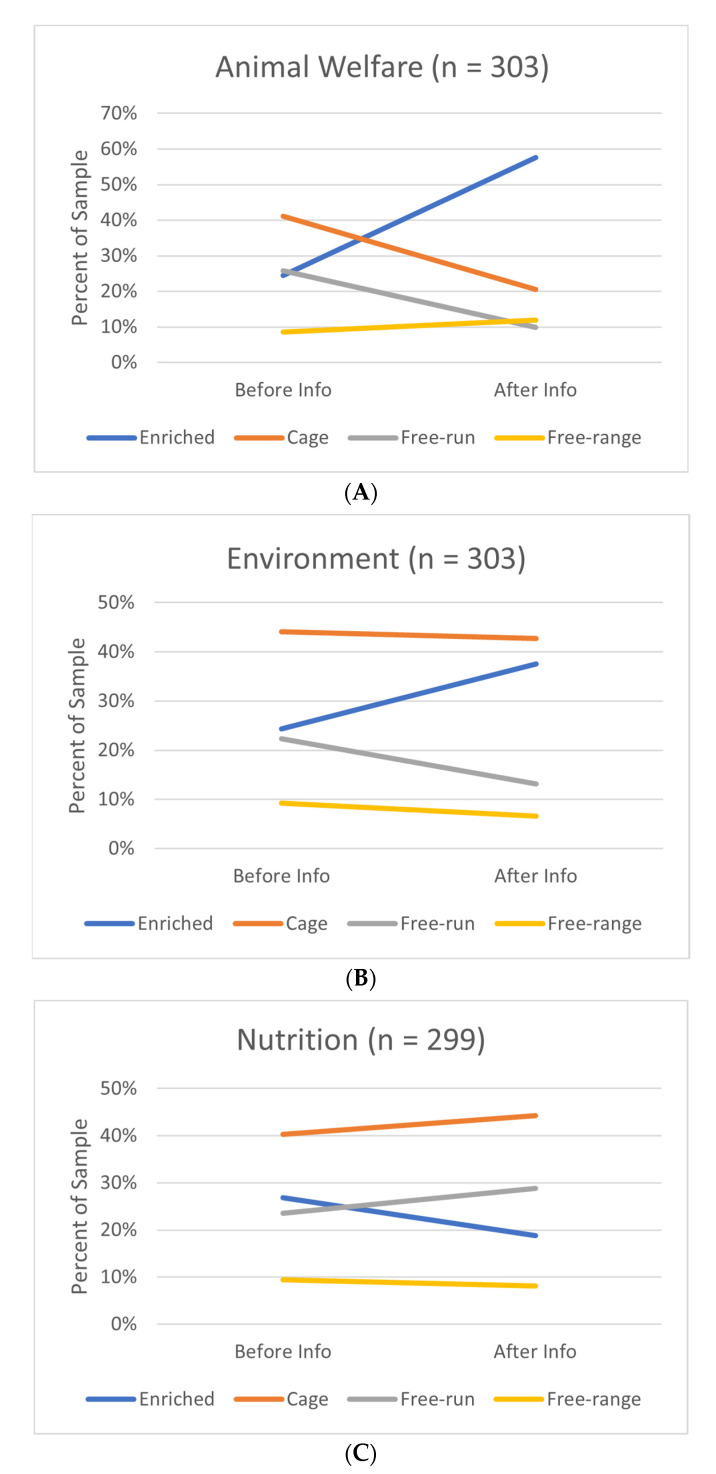
(**A**) Choice shares of each type of fresh egg before and after information—treatment animal welfare; (**B**) choice shares of each type of fresh egg before and after information—treatment environment; (**C**) choice shares of each type of fresh egg before and after information—treatment nutrition.

**Table 1 animals-13-00324-t001:** Price by type of eggs by price treatment.

	Cage	Enriched	Free Run	Free Range
Dozen eggs	CAD 3.59	CAD 4.10	CAD 5.89	CAD 6.10
PS1				
Salad	CAD 5.49	CAD 5.59	CAD 5.75	CAD 5.85
Quiche	CAD 7.99	CAD 8.39	CAD 9.49	CAD 9.99
PS2				
Salad	CAD 5.49	CAD 5.99	CAD 6.49	CAD 6.99
Quiche	CAD 7.99	CAD 8.09	CAD 8.19	CAD 8.29

**Table 2 animals-13-00324-t002:** Nutritional, animal welfare, and environmental scores attributed to each treatment in the choice sets per product. From best (A-green), (B-yellow), (C-orange) to worst (D-red).

	Egg Production System			
Treatment/Product	Cage	Enriched Housing	Free Run	Free Range
Nutrition				
Dozen Eggs	A	A	A	A
Quiche	C	C	C	C
Salad	B	B	B	B
Animal Welfare				
Dozen Eggs	D	A	C	B
Quiche	D	A	C	B
Salad	D	A	C	B
Environment				
Dozen Eggs	B	B	C	C
Quiche	A	A	B	B
Salad	A	A	B	B

**Table 3 animals-13-00324-t003:** Descriptive statistics of the sample and of the treatment subsamples.

	Full Sample		Animal Welfare		Environment		Nutrition	
	*n* = 905		*n* = 303		*n* = 303		*n* = 299	
Sex	*n*	%	*n*	%	*n*	%	*n*	%
Male	467	52%	144	48%	168	55%	155	52%
Female	438	48%	159	52%	135	45%	144	48%
Age								
18–34	213	24%	74	24%	73	24%	66	22%
35–54	412	46%	135	45%	141	47%	136	45%
55 and over	280	31%	94	31%	89	29%	97	32%
Role in household for buying food								
Main buyer	547	60%	169	56%	191	63%	187	63%
Shared responsibility	358	40%	134	44%	112	37%	112	37%
Frequency of egg consumption in household								
More than 8 dz/mth	35	4%	12	4%	17	6%	6	2%
From 4 to 8 dz/mth	137	15%	39	13%	50	17%	48	16%
From 1 to 3 dz/mth	575	64%	196	65%	182	60%	197	66%
Less than 1 dz/mth	158	17%	56	18%	54	18%	48	16%

**Table 4 animals-13-00324-t004:** Distribution of choices for a dozen eggs.

Production System (Price *)		
	*n*	%
Cage (CAD 3.59)	370	41%
Enriched (CAD 4.10)	220	24%
Free run (CAD 5.89)	217	24%
Free range (CAD 6.10)	98	11%

* For fresh eggs, the prices were identical in both price sets PS1 and PS2.

**Table 5 animals-13-00324-t005:** Individual patterns of choices from fresh eggs to a quiche or a salad with eggs.

	Egg-Quiche						Egg-Salad					
	LOW			HIGH			LOW			HIGH		
Transition in the Type of Egg	*n*	Group Size	%	*n*	Group Size	%	*n*	Group Size	%	*n*	Group Size	%
Same	252	453	56%	322	452	71%	235	452	52%	314	453	69%
More expensive	153	359	43%	85	390	22%	192	330	58%	90	398	23%
Less expensive	48	327	15%	45	281	16%	25	338	7%	49	292	17%

Group size = only individuals who can make the shift are included, hence individuals who purchase cage eggs, cannot downgrade, and are not included in the Downgrade group size, while individuals who purchase free-range eggs cannot upgrade and are therefore not included in the Upgrade group size. The same group includes all observation.

**Table 6 animals-13-00324-t006:** Changes in the choice of type of eggs for a dozen eggs before and after the information treatment.

	Animal Welfare		Environment		Nutrition	
	Before	After	Before	After	Before	After
Cage	41%	21%	44%	43%	40%	44%
Enriched	25%	58%	24%	38%	27%	19%
Free run	26%	10%	22%	13%	23%	29%
Free range	9%	12%	9%	7%	9%	8%

**Table 7 animals-13-00324-t007:** Change from the initial choice of type of eggs, by product, after nutrition score information.

	Eggs			Quiche			Salad		
Transitions in the Type of Egg	*n*	Group Size	%	*n*	Group Size	%	*n*	Group Size	%
No change	240	299	80%	241	299	81%	229	299	77%
Less expensive	33	175	19%	43	203	21%	47	212	22%
More expensive	26	270	10%	15	245	6%	23	243	9%

Group Size: ‘No change’ includes the full sample while the ‘Less expensive’ row excludes individuals in the least expensive choice to start with (cage) and the ‘More expensive’ row excludes individuals who started in the most expensive choice (free range).

**Table 8 animals-13-00324-t008:** Change from the initial choice of type of eggs, by product, after animal welfare score information.

	Eggs			Quiche			Salad		
Transitions	*n*	Group Size	%	*n*	Group Size	%	*n*	Group Size	%
No change (excl. enriched)	111	236	47%	105	240	44%	113	257	44%
From enriched to enriched ^1^	62	67	93%	49	63	78%	36	46	78%
Total no change	173	303	57%	154	303	51%	149	303	49%
To enriched with cost	57	127	45%	44	101	44%	38	97	39%
To enriched with saving	49	109	45%	71	139	51%	83	160	52%
Total to enriched	106	236	45%	115	240	48%	121	257	47%
Shifts not toward enriched									
Saving only (FAW loss)	3	184	2%	13	203	6%	8	212	4%
FAW gain at higher cost	17	207	8%	17	186	9%	18	201	9%
Lost of FAW with cost	4	67	6%	4	63	6%	7	46	15%

^1^ From Enriched to enriched considers respondents that chose enriched-type eggs before and after the information treatment. The no-change respondents exclude the respondents included in ‘From enriched to enriched’ (62), while the Total no change includes all respondents keeping the same choice as before the information treatment.

**Table 9 animals-13-00324-t009:** Change from the initial choice of type of eggs, by product, after environmental score (ES) information.

	Eggs			Quiche			Salad		
Transitions	*n*	Group Size	%	*n*	Group Size	%	*n*	Group Size	%
Initial is best ES-no change	158	201	79%	126	164	77%	121	148	82%
Initial is not best ES-no change	45	102	44%	62	139	45%	63	155	41%
Total no change	203	303	67%	188	303	62%	184	303	61%
Better ES with saving	43	102	42%	65	139	47%	73	155	47%
Same ES with saving	21	113	19%	17	108	16%	21	117	18%
Same ES with cost	23	190	12%	17	195	9%	16	186	9%
Lower ES with cost	13	201	6%	16	164	10%	9	148	6%

## Data Availability

The data presented in this study are available on request from the corresponding author. The data are not publicly available due to ethic committee restrictions on data privacy.

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
