# Peer review of "Do Consumers Value Welfare and Environmental Attributes in Egg Production Similarly in Fresh Eggs and Prepared Meals?"

_animals, 2023, doi:10.3390/ani13030324_

Round 1

Reviewer 1 Report

Line 21: Where/What is "Quebec consumers’" ? I know where it is, but most readers may not know, please add country information.
Line 35-37: Please add a reference
Line 50-55: Do not ask the research question in the middle of the introduction section. Instead, it would be better to leave an open-ended comment. Please rewrite this paragraph.
Line 65: boiler to broiler
Line 65-68: "As an example, in the production of broiler chickens; decreasing stocking density has a negative environmental impact (it increases net greenhouse gas (GHG) emissions per kilogram of chicken produced) but a positive social impact (it increases animal welfare)." Is there a reference for this comment? Stocking density=bird/square meter or total BW/m2. The decrease in stocking density is due to the decrease in broiler count or bw in the area. Either way, why increase GHG?
Line 91-92: "Note that the nutritional value of eggs is maintained constant in our experiment. Thus, housing system does not affect nutritional value." Do you think the nutritional value of the egg can be affected in any way by the rearing system?
Line 86-93: Some statements in this section are written informally, please do not write scientific articles in the form of chat.
Line 98: "nest boxes"
Line 122-127: I think it's an unnecessary explanation for the paper.
Line 236-238: Student t-test is any statistical hypothesis test in which the test statistic follows a Student's t-distribution under the null hypothesis. It is most commonly applied when the test statistic would follow a normal distribution if the value of a scaling term in the test statistic were known (typically, the scaling term is unknown and therefore a nuisance parameter). When the scaling term is estimated based on the data, the test statistic—under certain conditions—follows a Student's t distribution. The t-test's most common application is to test whether the means of two populations are different. So, unfortunately, it is not possible to do the parametric t-test with the data you collected through the survey.
Line 440 and later: You have mentioned many choice changes and their reasons. However, you did not specify whether these changes occurred according to the results of any hypothesis test. So how do you know there is a choice change? Appropriate statistical analysis results are required to make these comments. Please get support for statistical analysis.

Reviewer 2 Report

Your research is complex and not easy to present, although you have done it quite well. Perhaps the discussion in the "Results" section could have been omitted, and in the section "Discussion" the results should have been compared and/or explained using appropriate references, as is common. It would contribute to the quality of the work.
Additionally, as for the results, when mentioning statistical significance it is customary to indicate the p-value in parentheses (or at least the significance level, regardless of what was said in the "Methods" at which level it was determined).

I suggest giving the complete order of colors appearing in the tables: green (the best), yellow, orange, and red (the lowest score).

According to the instruction, the name of the section is "Material and Methods", not "Methodology". I suppose you should change that.

Other suggestions are related to minor technical errors (use of prepositions, plurals, commas, etc.), and you can find them in the comments provided in the reviewed manuscript.

Reviewer 3 Report

This paper shows an original investigation that may have great relevance from the point of view of the production sector, transformation industry, commercial distribution, hotel sector and egg consumers, although it lacks the approach and application of an analysis robust statistic.

The authors should improve the methodology by incorporating a more in-depth statistical analysis that allows validating the results obtained.

Round 2

Reviewer 1 Report

Dear Authors,

Thank you for meticulously following the specified parts.

Best regards,